Significance of TP53 mutation in bladder cancer disease progression and drug selection

Wu Guang
Wang Fei
Li Kai
Li Shugen
Zhao Chunchun
Fan Caibin fancaibin@sina.com
Wang Jianqing jqwang14@fudan.edu.cn
The Affiliated Suzhou Hospital of Nanjing Medical University , Suzhou , China
de Azevedo Jr Walter
Electronic publication date: 2019 Dec 16
Publication date: 2019
Volume: 7
Electronic Location ID: e8261
Received 2019 Sep 12; Accepted 2019 Nov 20
Copyright: © 2019 Wu et al.
Copyright year: 2019
Copyright holder: Wu et al.
License: This is an open access article distributed under the terms of the Creative Commons Attribution License, which permits unrestricted use, distribution, reproduction and adaptation in any medium and for any purpose provided that it is properly attributed. For attribution, the original author(s), title, publication source (PeerJ) and either DOI or URL of the article must be cited.
License URL: https://creativecommons.org/licenses/by/4.0/

Keywords: Bladder cancer, TP53 mutation, TCGA, RNA sequencing, Bioinformatics analysis, Drug selection

Funding: National Natural Science Foundation of China 81802565 Natural Science Foundation of Jiangsu Province BK20180216 Key Project of the Scientific Research Project of Nanjing Medical University Affiliated Suzhou Hospital szslyy2017005 This study was supported by National Natural Science Foundation of China (Grant No. 81802565); Natural Science Foundation of Jiangsu Province (Grant No. BK20180216); Key Project of the Scientific Research Project of Nanjing Medical University Affiliated Suzhou Hospital (Grant No. szslyy2017005). The funders had no role in study design, data collection and analysis, decision to publish, or preparation of the manuscript.

==============================
Background

The tumor protein p53 (TP53) mutant is one of the most frequent mutant genes in bladder cancer. In this study, we assessed the importance of the TP53 mutation in bladder cancer progression and drug selection, and identified potential pathways and core genes associated with the underlying mechanisms.

Methods

Gene expression data used in this study were downloaded from The Cancer Genome Atlas and cBioportal databases. Drug sensitivity data were obtained from the Genomics of Drug Sensitivity in Cancer. We did functional enrichment analysis by gene set enrichment analysis (GSEA) and the Database for Annotation, Visualization and Integrated Discovery (DAVID).

Results

We found the TP53 mutation in 50% of bladder cancer patients. Patients with the TP53 mutation were associated with a lower TP53 mRNA expression level, more advanced tumor stage and higher histologic grade. Three drugs, mitomycin-C, doxorubicin and gemcitabine, were especially more sensitive to bladder cancer with the TP53 mutation. As for the mechanisms, we identified 863 differentially expressed genes (DEGs). Functional enrichment analysis suggested that DEGs were primarily enriched in multiple metabolic progressions, chemical carcinogenesis and cancer related pathways. The protein–protein interaction network identified the top 10 hub genes. Our results have suggested the significance of TP53 mutation in disease progression and drug selection in bladder cancer, and identified multiple genes and pathways related in such program, offering novel basis for bladder cancer individualized treatment.

Introduction

Carcinoma of the urinary bladder (bladder cancer) is one of the worldwide most common malignancies, causing almost 150,000 deaths every year (Siegel, Miller & Jemal, 2016). Bladder cancer is prone to recurrence and has strong invasiveness, which means that about 25% of patients develop into muscle-invasive bladder cancer (MIBC) or metastatic disease as initial diagnosis or during the treatment and have a worse prognosis. Transurethral resection of the bladder (TURB) is the primary treatment of initial non muscle-invasive bladder cancer (NMIBC). After TURB, intravesical chemotherapy is the critical adjuvant therapy in all patients to prevent recurrence. Until now, surgery and chemotherapy are the primary treatments of MIBC. Today, individualized treatment, including molecular pathological diagnosis and individualized medicine testing, can help clinicians select target chemotherapy drugs according to the characteristics of patient’s tumor cells. Such individualized treatment could achieve better therapeutic effect, delay disease progression, and improve the prognosis. Therefore, to identify the feature of bladder cancer patients could provide candidate targets and strategies for individualized treatment.

Tumor protein p53 (TP53) is now identified as a tumor suppressor in various tumor types. It can regulate target gene expression in order to play multiple roles in coping with complicated cellular progression, resulting in transcriptional activation, senescence, apoptosis or changes in metabolism and cell cycle (Vousden & Prives, 2009). Mutation in TP53 is frequently found in human cancer, including bladder cancer. Previous studies have shown that nearly half of the MIBC samples had TP53 mutation, and that TP53 function was inactivated in 76% of samples (The Cancer Genome Atlas Research Network, 2014; Rentsch et al., 2017). The TP53 mutation and TP53 related pathways could serve as driver mutations in bladder cancer, which promotes disease progression and influences cancer prognosis and therapeutic strategy (Cazier et al., 2014; Lang et al., 2004). Mutant TP53 also induces the activation of mechanisms in cancer initiation and progression, which contributes to poor disease outcome (Kim, Zhang & Lozano, 2015; Smith et al., 2002). Many researchers have shown that mutant TP53 could accelerate metastatic tumor cell proliferation and promote the metastatic ability (Morton et al., 2010). Moreover, TP53 status has been shown to influence chemotherapy and drug sensitivity in bladder cancer (Pandey, Bourn & Cekanova, 2018; Ye et al., 1998). As TP53 mutation is the most frequently detected mutation in bladder cancer patients (Lamy et al., 2006), we intend to study whether clustering analysis of data from molecular profiling would define TP53 mutant bladder tumors as a subgroup to help identify some novel individualized treatment methods.

In the present study, we explored the Genomics of Drug Sensitivity in Cancer (GDSC) database to estimate the significance of TP53 mutation in bladder cancer drug selection, and analyzed RNA sequencing (RNA-Seq) dataset of MIBC to find out the role of TP53 mutation in disease progression, providing novel individualized treatment options. Meanwhile, the identification of critical pathways and core genes associated with TP53 mutation could uncover the potential mechanisms and therapeutic targets.

Materials and Methods

Analysis of the GDSC database

Genomics of Drug Sensitivity in Cancer project is setup for cancer molecular therapeutics and mutation exploration (Vanden Heuvel et al., 2018). Public online platform has been developed to help researchers to analyze the astronomical data matrix and download data (Yang et al., 2013). In this article, we first searched compounds with significant selectivity for TP53 mutation, and then tried to narrow the hits by check the sensitivity only in bladder cancer cells. The volcano plots, scatter plots, and the statistical analysis were downloaded directly from the GDSC online platform.

Gene set enrichment analysis

We analyzed gene set enrichment analysis (GSEA) to find out the differences in gene mRNA expression levels of biological annotation and pathways to help understand the effect of TP53 mutation on biological function gene sets in bladder cancer patients.

Identification of differentially expressed genes

We used EdgeR to examine differential gene expression between TP53 mutation and wild-type bladder cancer patients (McCarthy, Chen & Smyth, 2012; Robinson, McCarthy & Smyth, 2010). Differentially expressed genes (DEGs) were identified with the following criterion: |fold change| ≥ 2; P-value < 0.05.

Functional annotation and pathway enrichment analysis of DEGs

Here, we used the Database for Annotation, Visualization and Integrated Discovery (DAVID) to complete gene functional analysis (Dennis et al., 2003). Gene Ontology (GO) and Kyoto Encyclopedia of Genes and Genomes (KEGG) analysis were uploaded to DAVID website (https://david.ncifcrf.gov/) for enrichment analysis, while specifying a P-value < 0.05 for statistical significance.

Statistical analysis

We used the Student’s t-test to compare the mRNA expression level in different TP53 status bladder cancer tissue. The association between TP53 mutation and the clinic parameters were evaluated by a χ2 test. The Kaplan–Meier method with log-rank test was used for calculating the clinical prognosis between by Graphpad. In GSEA, enrichment results satisfying a nominal P-value cutoff of 0.05 with a false discovery rate (FDR) q-value < 0.25 were considered statistically significant. All the statistical analyses were conducted with R 3.3.0 and Graphpad. A value of P < 0.05 was considered statistically significant.

Results

TP53 mutation is more common in MIBC

We first downloaded the information and complete follow-up profiles of 408 MIBC patients and their cancer tissue expression data from The Cancer Genome Atlas (TCGA) database. There were 206 patients (49.5%) with TP53 mutation (Fig. 1A). Main mutation types were amplification, truncating, deep deletion, inframe mutation and missense mutations spanning over entire gene (Fig. 1B). To initially explore whether TP53 mutation is an important factor in the progression of bladder cancer, we also analyzed the proportion of TP53 mutation in NMIBC. However, there were only 21% patients with less TP53 mutation types in another dataset of NMIBC, indicating the possibility that TP53 mutation might contribute to the progression to muscle-invasive disease, and that TP53 mutation might be a potential mark of bladder cancer being easy invasiveness (Figs. 1C and 1D).

Figure 1 Mutation frequency and types of TP53 in bladder cancer from the cancer Genome Atlas (TCGA) database.

(A) Mutation frequency of TP53 in MIBC. (B) TP53 protein mutation diagram showed the mutation types of TP53 in MIBC. P53_TAD: P53 transactivation motif, P53_tetramer: P53 tetramerisation motif. (C) Mutation frequency of TP53 in NMIBC. (D) TP53 protein mutation diagram showed the mutation types of TP53 in NMIBC. P53_TAD: P53 transactivation motif, P53_tetramer: P53 tetramerisation motif.

TP53 mutation in bladder cancer progress and prognosis

We next explored the influence of TP53 mutation on bladder cancer development and prognosis. We first compared the clinical information of bladder cancer patients in the two groups. After analyzing, we found that patients with TP53 mutation have higher neoplasm histologic grade and more advanced tumor stage, indicating that TP53 mutation might contribute to the disease progression (Table 1).

Table 1 Clinical Characteristics of BLCA Patients and TP53 status in TCGA.

Characteristics	TP53 status	P value	
Wild type	Mutated	
Mean Age, years	67.54	68.75		
Range	34–90	44–90		
Gender			P = 0.7853	
Female	54	53		
Male	156	144		
Tumor T stage			P = 0.6904	
T1	2	2		
T2	23	14		
T2a	15	11		
T2b	30	26		
T3	23	19		
T3a	30	41		
T3b	42	39		
T4	4	7		
T4a	25	18		
T4b	2	2		
N stage			P = 0.0166	
N0	136	100		
N1	22	24		
N2	34	41		
N3	1	7		
NX	14	22		
M stage			P = 0.0314	
M0	112	79		
M1	5	6		
MX	92	110		
AJCC neoplasm disease stage			P = 0.1580	
Stage I	2	0		
Stage II	69	60		
Stage III	77	63		
Stage IV	60	74		
Neoplasm histologic grade			P = 0.0013	
High	190	193		
Low	18	3		
Note:

P < 0.05 was considered statistically significant. Chi-square test was used.

We then investigated TP53 mRNA expression in TP53 wild type and mutated groups. Results indicated that TP53 expression level was lower in mutated bladder cancer patients’ tissue (Fig. 2A). However, TP53 mutation had no effect on disease prognosis of survive (Fig. 2B) and recurrence (Fig. 2C). Results above indicated that TP53 mutation may contribute to bladder cancer disease progression, but not the prognosis. Early intervention may be benefit to such patients.

Figure 2 TP53 mutation and bladder cancer prognosis.

(A) Correlation between TP53 mutation and mRNA expression. (B & C) Kaplan–Meier survival and disease recurrence curves for bladder cancer patients stratified by TP53 mutation.

Bladder cancer cells with TP53 mutation are sensitive to mitomycin-C, doxorubicin and gemcitabine

Chemotherapy is one of the chief treatments for bladder cancer. Therefore, the sensitivity to chemotherapy drugs determines the therapeutic effect. In addition to TP53 mutation on cancer progression, we also investigated whether TP53 mutation induces drug resistance in bladder cancer. In order to explore the role of TP53 mutation in drug sensitivity and find out specific inhibitors for individualized treatment, we explored the GDSC database to investigate whether TP53 mutated patients have selective compounds in bladder cancer. Results indicated that bladder cancer cells that harbor TP53 mutation, not other types of tumors, were significant sensitive to mitomycin-C, doxorubicin and gemcitabine (Fig. 3), making them potential target compounds. As a result, mitomycin-C, doxorubicin and gemcitabine conferred potential individualized compounds for bladder cancer patients with TP53 mutation.

Figure 3 TP53 mutation influences drug selection of bladder cancer.

(A) Volcano plotting showed that bladder cancer with TP53 mutation was significantly sensitive to mitomycin-c, doxorubicin and gemcitabine. (B–G) Reproduction of GDSC database showed that bladder cancer cells with TP53 mutation, but not cancer of other types, was significantly inhibited by Mitomycin-C, Doxorubicin and Gemcitabine. NS: not significant, * P < 0.05, ** P < 0.01, *** P < 0.001.

Results of GSEA

Our results above showed that TP53 mutation plays an important role in bladder cancer disease development and drug selection. Then we analyzed the effects of TP53 mutation on cellular process to investigate the underlying mechanism. We first used GSEA approach to analyze biological functional gene sets. GSEA is a method to reveal the metabolic pathway enrichment situation of all the gene expression of the selected sample for enrichment analysis. Enrichment results satisfying a nominal P-value cutoff of 0.05 with a FDR q-value < 0.25 were considered statistically significant in the present study. We found 23 gene sets that were significantly enriched (Table S1) and showed the 16 most enriched gene sets obtained by normalized enrichment score value ranking in Fig. 4. Among the 16 gene sets, DNA repair, G2M checkpoint, glycolysis, PI3K-AKT-mTOR signaling, MYC targets, mTORC1 signaling, cholesterol homeostasis, protein secretion, peroxisome, E2F targets, androgen response, reactive oxygen species (ROS) pathway and mitotic spindle are closely related to tumorigenesis. These results suggest that TP53 mutation may promote bladder cancer progression by influencing pathways in cancer, ROS, metabolism and DNA repair.

Figure 4 GSEA results of TP53 mutation in bladder cancer patients.

Including DNA repair (A), G2M check point (B), glycolysis (C), PI3K-AKT-mTOR signaling pathway (D), MYC target (E), spermatogenesis (F), mTORC1 signaling pathway (G), heme metabolism (H), UV response (I), cholesterol homeostasis (J), protein secretion (K), peroxisome (L), E2F targets (M), androgen response (N), reactive oxygen species pathway (O), and mitotic spindle (P).

Identification of DEGs

We next identified the DEGs to find out the pathways and genes implicated in TP53 mutation. Based on the in silico analysis, 863 genes were identified as DEGs, among which 678 were upregulated and 185 were downregulated. The volcano plot of the DEGs is shown in Fig. 5A.

Figure 5 DAVID enrichment results of differentially expressed genes.

(A) Volcano plot for differentially expressed genes. (B) GO enrichment terms of differentially expressed genes. (C) KEGG pathway analysis of differentially expressed genes.

GO and KEGG analysis of DEGs

Then we uploaded the 863 DEGs online for further functional enrichment analyses using DAVID. Results of GO analysis (Fig. 5B) suggested significant enrichment in positive regulation of immune response, calcium ion-regulated exocytosis of neurotransmitter, regulation of ion transmembrane transport, regulation of membrane potential, adenylate cyclase-activating adrenergic receptor signaling pathway, xenobiotic metabolic process, acute-phase response, detection of chemical stimulus involved in sensory perception of smell, angiotensin maturation, adult walking behavior, layer formation in cerebral cortex, regulation of calcium ion-dependent exocytosis and G-protein coupled receptor signaling pathway.

Moreover, KEGG pathway analysis showed significant enrichment in chemical carcinogenesis, metabolism of xenobiotics by cytochrome P450, retinol metabolism, drug metabolism—cytochrome P450, neuroactive ligand-receptor interaction, steroid hormone biosynthesis, olfactory transduction, glioma, transcriptional misregulation in cancer, cytokine–cytokine receptor interaction, bladder cancer, gastric acid secretion and Jak-Stat signaling pathway (Fig. 5C).

We then used the Search Tool for the Retrieval of Interacting Genes database to investigate the interaction and hub genes of DEGs. The top 10 genes ranked by degree were identified as hub genes, including epidermal growth factor receptor (EGFR), GNG4, PRKACG, CSF2, CTAG1A, CTAG1B, BAGE5, GABBR2, F2, PVALB. EGFR had the highest degree of nodes among the hub genes with 26.

Discussion

Tumor protein p53 acts as a lipid phosphatase, removing the phosphate in the D3 position of the inositol ring. Previous studies showed that the major function of TP53 relies on its phosphatase activity. TP53 showed antagonism of the PI3K/AKT pathway, while AKT-independent function was also found in other studies (Maehama, Taylor & Dixon, 2001). TP53 mutation is frequently found in multiple human cancers, including bladder cancer. TP53 mutation and related pathways could be drivers in bladder cancer initiation (Cazier et al., 2014; Williamson, Elder & Knowles, 1994). TP53 mutation and other critical gene mutations have been shown to alter multiple cancer related pathways in bladder cancer and thus promote disease progression, which makes it a potential therapeutic target (Bakkar et al., 2003; Choi et al., 2014; Gui et al., 2011; Liu & Kwiatkowski, 2015). Here in the present study, we evaluated the clinical significance of TP53 mutation in bladder cancer to provide some basis for individualized treatment. Mechanisms of TP53 mutation promoting disease progression were also analyzed.

In clinical terms, patients with MIBC have more TP53 mutations than patients with NMIBC (Fig. 1). Clinical characteristic analysis of bladder cancer patients in both groups showed higher neoplasm histologic grade and more advanced tumor stage in bladder cancer patients with TP53 mutation (Table 1). Results above suggest the role of TP53 mutation as an indicator to advanced tumor with malignant potential, and that TP53 mutation may be involved in the progression of bladder cancer. In addition, detection of tumor gene mutations now can help clinicians judge the prognosis of patients after surgery and recommend better individualized treatment strategies. Our results showed the evidence that bladder cancer patients with TP53 mutation might need early and comprehensive intervention to live longer. Especially, bladder cancer patients with TP53 mutation may require early systemic chemotherapy, other than merely intravesical chemotherapy, to cope with disease progression. Besides, data from GDSC show preliminary evidence that mitomycin-C, doxorubicin and gemcitabine exhibit sensitivity for bladder cancer cells that harbor TP53 mutation, which provides evidence for the use of specific anti-tumor drugs to such patient in clinical practice (Fig. 3). Mitomycin-C, doxorubicin and gemcitabine are the first line chemotherap drugs, which could decrease cell proliferation and metastasis in human bladder cancer cell lines and have been shown to decrease the number of subsequent recurrences and increase the recurrence-free interval in bladder cancer by multiple mechanisms (such as hyperthermia) (Galsky et al., 2015; Humphrey & Mann, 1949; Sternberg, 2000; Tolley et al., 1996; Van Der Heijden et al., 2004; Von Der Maase et al., 2000; Yu et al., 2018). The effects of mitomycin-C, doxorubicin and gemcitabine on cells with or without mutated TP53 have been well established. Mitocmycin-C has been reported to induce TP53 independent cell death and the expression of TP53 was not associated with the outcomes of patient treated with mitomycin-C (Seo et al., 2010). Early preliminary researches have indicated that doxorubicin has TP53-dependent cell death response and TP53 mutantin could enhance chemosensitivity of doxorubicin in bladder cancer cells (Bilim et al., 2000; Chang & Lai, 2001), while gemcitabine is effective in bladder cell lines independent of TP53 status (Fechner et al., 2003). However, some results are still controversial and various confounding factors could influence drug sensitivity. MDM2 is an oncogene that acts by suppressing TP53 function in multiple cancer types (Toi et al., 1997). MDM2 overexpression has been found in some bladder cancer cells and contributes to doxorubicin resistance in bladder cancer, which could be a confounding factor to conclude that doxorubicin would be a suitable drug to target tumors with mutant TP53 (Shiina et al., 1999; Smith et al., 2003). The expression of pro-oncogenic RUNX2 could also confuse the results of gemcitabine treatment in cancer with TP53 mutation (Ozaki et al., 2016, 2018). Besides, mutant TP53 could produce a stable protein, so further analyses would focus on the protein level to conclude more accurate results. To sum up, our results could provide some novel preliminary evidence in developing better individualized treatment strategies. Until now, the research data of relationship between TP53 mutation and bladder cancer chemosensitivity are still limited and the final conclusion is still controversial owing to confounding factors. Because of the critical role of chemotherapy in bladder cancer treatment, further basic and clinical investigations are necessary to determine the role of TP53 mutation in chemotherapy and drug selection.

To investigate the underlying mechanisms, we analyzed the gene expression data of MIBC to identify the key pathways and core genes associated with TP53 mutation. In the present study, we combined the results of GO/KEGG and GSEA to have the final conclusion. The input variable of GSEA is the gene expression level, and the input variable of GO/KEGG pathway enrichment analysis is the gene list. Both analyses can screen out the significantly enriched pathway, the difference is that GSEA is for all genes, GO/KEGG is for differential gene enrichment. Results of GSEA analysis suggest that TP53 mutation was mainly associated with multiple cancer related pathways, cell proliferation and division, DNA repair, ROS and metabolism (Fig. 4). All programs above are all important contributors to bladder cancer initiation and development. Excessive accumulation of ROS could be a byproduct intracellular by mitochondria and other cellular elements and exogenously by pollutants, tobacco, smoke, drugs, xenobiotics, and radiation, thus promoting cancer through modulating various cell signaling pathways and inducing epigenetic changes in genes in various cancer types, including bladder cancer (Prasad, Gupta & Tyagi, 2017). Metastatic bladder cancer cells show different antioxidant expression profile, resulting in an increase in ROS production network (Hempel et al., 2009; Miyajima et al., 1997). As TP53 is one of the inhibitor of PI3K-AKT-mTOR signaling pathway, TP53 mutation influences the mTORC1 signaling pathway activity definitely.

We next identified the DEGs and carried out the functional enrichment analyses (Fig. 5). Results show 863 genes as DEGs with different TP53 mutation status. Enrichment analysis suggested that DEGs in TP53 mutation bladder cancer patients were related to multiple cellular programs. Among all programs, we found chemical carcinogenesis as the top KEGG term. As we all know, the occurrence of bladder cancer is closely related to the long-term stimulation of bladder cells by chemical carcinogens. Our results indicate the possibility that TP53 mutation might aggravate the carcinogenic effects of various carcinogens on bladder cells. On the other hand, it is very likely that it is the large number of chemical carcinogens that stimulate and induce mutations in TP53. The results of KEGG/GO combined with GSEA and the clinical impact of TP53 on the disease progression highlights the importance of early prevention of bladder cancer, and the significant role of metabolism and cancer related pathways in bladder cancer development.

In PPI network analysis, we found out the top 10 hub genes with the highest degree of interaction. Amplification and mutations of EGFR have been shown to be driving events in several solid tumors including bladder cancer (Villares et al., 2007). Also, bladder cancer cells are sensitive to treatment with drugs that targeting the EGFR pathway, which has been shown to be a therapeutic target in dealing with advanced bladder cancer (Mason et al., 2009).

Our study contained one limitation. In our study, we can only compare the mutation frequency and type of TP53 in MIBC and NMIBC. Because of the lack of clinical characteristics, prognosis and gene expression data of TP53 mutated NMIBC in TCGA, we failed to further compare these attributes. To further distinguish these different types of bladder cancer needs more data and further study.

Conclusions

In conclusion, this study found out the clinical significance of TP53 mutation in bladder cancer, and the main pathways and genes associated with TP53 mutation, which may facilitate developing early intervention and providing better therapeutic strategies against such special subtype of bladder cancer. Furthermore, the mechanism and validation of TP53 mutation in bladder cancer still need further research in clinical and molecular biology experiments.

Supplemental Information

Supplemental Information 1 Results of GSEA in TP53 mutated bladder cancer.

Click here for additional data file.

We acknowledge the cBioPortal for Cancer Genomics site and the TCGA Research Network for generating TCGA datasets.

Additional Information and Declarations

Competing Interests

Author Contributions

Data Availability

The authors declare that they have no competing interests.

Guang Wu performed the experiments, analyzed the data, prepared figures and/or tables, approved the final draft.

Fei Wang performed the experiments, authored or reviewed drafts of the paper, approved the final draft.

Kai Li performed the experiments, analyzed the data, prepared figures and/or tables, approved the final draft.

Shugen Li analyzed the data, contributed reagents/materials/analysis tools, prepared figures and/or tables, approved the final draft.

Chunchun Zhao analyzed the data, contributed reagents/materials/analysis tools, authored or reviewed drafts of the paper, approved the final draft.

Caibin Fan conceived and designed the experiments, authored or reviewed drafts of the paper, approved the final draft.

Jianqing Wang conceived and designed the experiments, performed the experiments, contributed reagents/materials/analysis tools, prepared figures and/or tables, authored or reviewed drafts of the paper, approved the final draft.

The following information was supplied regarding data availability: The MIBC RNA-Seq dataset was downloaded directly from TCGA database (accession number: TCGA-BLCA; https://portal.gdc.cancer.gov/projects/TCGA-BLCA). The corresponding clinic information was obtained from the cBioPortal for Cancer Genomics website (Bladder Cancer; TCGA, Cell 2017; http://www.cbioportal.org/study/summary?id=blca_tcga_pub_2017). The mutation information of NMIBC was also obtained from the cBioPortal for Cancer Genomics website (Nonmuscle Invasive Bladder Cancer; MSK Eur Urol 2017; http://www.cbioportal.org/study/summary?id=blca_nmibc_2017). DOI 10.1158/2159-8290.CD-12-0095.

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
