# Peer review of "Significance of TP53 mutation in bladder cancer disease progression and drug selection"

_PeerJ, doi:10.7717/peerj.8261_

## Round 0.1 · original submission · Major Revisions

The manuscript has been evaluated by three specialists in the field and they recommend revisions. English language revision is requested by two reviewers.

Reviewer 1 ·

Basic reporting

There are no issues with the basic structure of the manuscript.

However, there are issues with language usage throughout, particularly incorrect word use (e.g. line 132, "scope"; line 76, "astronomical"), lack of definite/indefinite articles, incorrect word forms, & singular/plural disagreement. Use of informal language, such as "dig out", "nowadays", & "what's more" should be avoided, as well. There are also spelling (e.g. line 146, "oxigen"; line 229, "cacner") & punctuation (e.g. line 164, ",.") errors in need of correction.

Citation of appropriate background literature early in both the Introduction & Discussion is inadequate, as well. It would also be helpful to cite the appropriate Figs. when Results are directly referenced in the Discussion (e.g. lines 205-226).

Finally, abbreviations for "GO" & "KEGG" analysis need to be defined.

Experimental design

The study appears to be within the scope of the journal. Its hypothesis & methodology are also adequately described.

However, the authors missed a huge opportunity to further compare MIBC vs. NMIBC disease patients. After Fig. 1 introduced a very interesting, significant(?) difference in their TP53 mutation frequencies, the remainder of the paper does not distinguish these different types of bladder cancer at all. This is a major weakness of their study.

Validity of the findings

The biggest issue with this manuscript lies in the poor presentation & discussion of its findings, which prevents assessment of their true validity.

Fig. 1. The 3 panels of this Fig. are not explained nearly well enough in either the legend or Results section. For example, the color coding in the odd, single bars of panels A & C is very difficult to read. The mutation frequencies should be associated with numerical values on an axis, not just a qualitative bar chart. Labeling panel A = 50% & panel C = 21% does not provide any true quantitation. There is also no key to the color coded regions of panel B. For example, what is "P53_tetr_"? This is apparently a TP53 gene/protein(?) diagram, although this is not explicitly stated anywhere. Finally, why wasn't another TP53 gene/protein diagram generated for the NMIBC data in panel C?

Table 1. "Age" should be "Mean Age"? Do P-values refer to the Stage/Grade totals? If so, then those total values should be provided. If not, how were they calculated?

Fig. 2. The P-values are not clearly associated with panels B & C. The color keys labeled generically as "....Query Genes" need to be edited to show "Wild-type" or "Mutant".

Fig. 4. This data requires much more work on both its explanation & presentation. The analyses for 16 'functional gene sets' are shown in Fig. 4, while the Results state that 13 were significantly enriched. What about the other 3 sets in Fig. 4? Furthermore, the enriched 'mTORC1 signaling' gene set mentioned in the Results does not even appear to be shown in Fig. 4. In addition, there is no way that readers unfamiliar with this particular analysis could interpret this data without far more explanation of its presentation. Most importantly, what are the criteria used to establish 'significant enrichment'? As presented, the 16 traces in Fig. 4 appear to be quite similar. It may be helpful to show an example of a gene set that was not significantly enriched. If some of the 16 shown are not significant, then they should be labeled as such & separated from the significant sets.

Fig. 5. How do the 'functional gene sets' presented in Fig. 4 mesh with the DEG's identified in Fig. 5? This needs to be addressed explicitly in the Discussion. As it is, these results don't appear to be linked in any useful context.

·

Basic reporting

The paper by Wu and colleagues is clearly written and understandable. Since TP53 is the most frequent mutated gene in human tumors it is very relevant.

Experimental design

The experimental design seems reasonable. However more details would be helpful to learn how the GDSC database concerning drug sensitivity was searched.

Validity of the findings

No comment.

Additional comments

The discussion should contain a section dealing with relevant clinical studies. It is essential to know if clinical studies with Mitomycin, Doxorubicin or Gemcitabine have been performed and how the outcome was. This should also be put in the context of TP53 mutation.

Reviewer 3 ·

Basic reporting

English editing is recommended.
The literature references were insufficient. Other groups have reported p53 mutations and p53 related pathways as driver mutations in bladder cancer: for example Cazier et. al. (2014), Nature Communications, and this paper and others were not referenced.

Experimental design

The experimental design and methods were appropriate. The novelty of the study was not obvious, and much of the relevant p53 related literature that has published similar findings was not cited.

Validity of the findings

The p53 literature required more careful consideration and many of the study's findings have been published or would be expected based on the literature. The drug related aspects of the study could be relevant to treatment of bladder cancer patients with mutant p53, but a more careful analysis of other confounding factors is required. The affect mitomycin-c, doxorubicin and gemcitabine has on cells with and without mutant p53 is reasonably well established, but again this literature was poorly cited. Mitocmycin-c has been reported to induce p53 independent cell death. Doxorubicin largely has a p53 dependent cell death response, some bladder cancers have enhanced MDM2 expression and this could be a confounding factor to conclude that doxorubicin would be a suitable drug to target tumors with mutant p53. The expression of RUNX2 could confound the results related to p53 mutations and gemcitabine treatment. Wild-type p53 is stabilized at the protein level and many p53 mutations produce a stable protein, so the analyses related to TP53 expression are limited without an analysis of p53 protein levels, or the expression of downstream targets of p53 such as CDKN1A.

---

## Round 0.2 · accepted · Accept

In my view, the authors carried out substantial modifications in the text, and the manuscript improved a great deal. It can be accepted as it is.